# Electrochemically Oxidized Carbon Nanotube Sheets for High-Performance and Flexible-Film Supercapacitors

**DOI:** 10.3390/nano13202814

**Published:** 2023-10-23

**Authors:** Jun Ho Noh, Jimin Choi, Hyunji Seo, Juwan Kim, Changsoon Choi

**Affiliations:** 1Department of Energy and Materials Engineering, Dongguk University, 30 Pildong-ro, 1-gil, Jung-gu, Seoul 04620, Republic of Korea; shwnsgh15@dongguk.edu (J.H.N.); james010828@gmail.com (J.C.); amy0527330@gmail.com (H.S.); kjw3328@dungguk.edu (J.K.); 2Department of Advanced Battery Convergence Engineering, Dongguk University, 30 Pildong-ro, 1-gil, Jung-gu, Seoul 04620, Republic of Korea

**Keywords:** electrochemical oxidation, carbon nanotube, supercapacitors, flexibility, wearable

## Abstract

The development of flexible, high-performance supercapacitors has been a focal point in energy storage research. While carbon nanotube (CNT) sheets offer promising mechanical and electrical properties, their low electrical double-layer capacitance significantly limits their practicability. Herein, we introduce a novel approach to address this challenge via the electrochemical oxidation treatment of CNT sheets stacked on a polyethylene terephthalate substrate. This introduces oxygen-containing functional groups onto the CNT surface, thereby dramatically enhancing the pseudocapacitive effect and improving ion adsorption. Consequently, using the material in a two-electrode system increased the capacitance by 54 times compared to pristine CNT. The results of electrochemical performance characterization, including cyclic voltammograms, galvanostatic charge/discharge curves, and capacitance retention testing data, confirm the efficacy of the electrochemical oxidation approach. Furthermore, the mechanical flexibility of the electrochemically wetted CNT sheets was validated through resistance and discharge retention testing under repetitive bending (98% capacitance retention after 1000 bending cycles). The results demonstrate that electrochemically wetted CNT sheets retain their intrinsic mechanical and electrical properties while significantly enhancing the electrochemical performance (0.59 mF/cm^2^ or 97.8 F/g). This work represents a significant advancement in the development of flexible, high-performance supercapacitors with potential applicability to wearable electronics, flexible displays, and next-generation energy storage solutions.

## 1. Introduction

The development of efficient and sustainable energy storage systems is a critical challenge in modern society [1,2,3,4]. Among the various pertinent materials, carbon nanotube (CNT) sheets have emerged as a promising candidate for supercapacitor electrodes due to their unique combination of mechanical strength, electrical conductivity, and high surface area [5,6,7,8,9,10]. These properties make them especially apt for applications requiring flexibility and durability, such as wearable electronics and portable energy storage devices [6,10,11,12,13,14,15]. Indeed, numerous studies have confirmed the efficacy of flexible supercapacitors using CNT sheet-based electrodes drawn from CNT forests [8,14]. More importantly, forest-spinnable CNT sheets are a more advantageous electrode material than classical powder CNTs. Anisotropic alignment in the drawing direction and among the highly aligned bundled structures with branches between them results in a networked microstructure with excellent electrical, mechanical, and electromechanical properties [16]. However, the inherent low-charge storage capability of CNT sheets impedes their widespread use in supercapacitors. Given that normalized capacitance is vital for supercapacitor energy storage applications, enhancing capacitance without degrading other CNT sheet properties is a pressing research concern. The low capacitance of CNT sheets can be attributed to two main factors: their innate hydrophobicity, which limits ion adsorption in aqueous electrolytes and consequently restricts the charge storage capacity [17,18,19,20], and their charge storage mechanism, which in this case is an electrical double-layer capacitance with typically lower capacitance than pseudocapacitance [21,22]. This limitation significantly impedes harnessing the full potential of CNT sheets in high-performance supercapacitors.

Various methods have been explored to address this limitation, such as doping with heteroatoms or coating with conductive polymers [23,24,25]. However, these approaches often compromise the intrinsic properties of CNTs, including their mechanical strength and electrical conductivity [26]. Moreover, these methods can complicate the fabrication process, thus making it less scalable and cost-effective. Another approach is to introduce functional groups onto the surface of CNTs. However, the chemical functionalization of the inactive CNT surface frequently introduces sp3 defects that disrupt π-π conjugation and, thus, decreases the inherent mechanical properties, thermal stability, and electrical conductivity of the CNTs [27,28,29]. Therefore, creating CNT sheets with high charge storage capability while maintaining their mechanical properties or deformability would make it possible to fabricate flexible, practicable power sources. From this perspective, the proposed electrochemical oxidation treatment (EOT) of CNT sheets could improve the capacitance while avoiding the use of brittle pseudocapacitive materials by appropriately controlling surface oxidation.

To overcome these challenges, we introduce a groundbreaking approach to significantly enhance the capacitance of CNT sheets without compromising their intrinsic properties. We employed an electrochemical activation method commonly known as EOT to fully activate the surfaces of CNT sheets stacked on a polyethylene terephthalate (PET) substrate. The EOT process involves the application of an electrical potential to the CNT sheets submerged in an electrolytic solution to facilitate the introduction of oxygen-containing functional groups, such as carboxyl, hydroxyl, and epoxy groups, onto the surface of the CNTs. These functional groups significantly improve the CNT surface’s wettability, thereby enhancing ion adsorption and electrochemical performance [30,31,32]. Remarkably, EOT achieves this without degrading the intrinsic properties of the CNT sheets while preserving their mechanical and electrical attributes, thus ensuring their usability in flexible supercapacitors [33,34,35]. This is possible because the degree of oxidation can be largely adjusted by controlling the oxidation time or applied voltage. Our method enables mild oxidation to the CNT sheets, which minimizes the disruption of π-π conjugation. Thereby, we produced a thin film-shaped supercapacitor with both highly improved capacitance (54-fold) compared to untreated CNT sheets and high mechanical flexibility (180° bendability).

While our group has previously used electrochemical activation to oxidize CNT yarns for supercapacitors [23,33] and actuators [17], the impact of EOT on CNT sheets remains unexplored. By employing EOT, our research not only addresses the critical limitation of low capacitance in pristine CNT sheets but also paves the way for their applicability to thin, flexible, high-performance supercapacitors. This scalable and efficient approach to functionalizing CNT sheets represents a significant breakthrough in the field of advanced energy storage applications.

## 2. Materials and Methods

### 2.1. Chemicals and Materials

CNT sheets were drawn from well-aligned multiwalled CNT forests with a height of 750 μm grown via chemical vapor deposition (A-Tech System Co., Incheon, Republic of Korea). Silver paste (CANS, Tokyo, Japan) and Sil Poxy (Smooth-on Inc., Macungie, PA, USA) were used to construct a CNT film supercapacitor cell for use in the electrochemical analysis. Sodium sulfate (Na_2_SO_4_) and polyvinyl alcohol (PVA; average molecular weight = 130,000 Da) were used to prepare the gel electrolyte (Sigma-Aldrich, St. Louis, MO, USA). PET Film (Film Bank, Kyeongkido, Republic of Korea) was used as the CNT sheet supercapacitor substrate.

### 2.2. Preparation of EOT CNT Sheets and the Gel Electrolyte

Five layers of CNT sheets (10 mm width and length) were drawn from a CNT forest and then carefully stacked on a 5 mm thick PET substrate. Electrical interfaces (essential for the EOT process) were established by affixing copper wires to both ends of the multi-layered CNT film. Silver paste was generously applied over the copper leads (diameter 180 μm) to maintain electrical connectivity. A non-brittle Sil Poxy layer was applied over the silver paste to ensure the bending stability of the EFAS. EOT was carried out using a three-electrode system with the pristine CNT film supercapacitor, Ag/AgCl, and platinum mesh as the working, reference, and counter electrodes, respectively. A 0.1 M aqueous Na_2_SO_4_ solution was utilized as the ionic medium. Subsequently, the appropriate electrochemical potential range relative to the Ag/AgCl reference electrode was administered via an electrochemical analysis system. To quantify the EOT effect, cyclic voltammetry (CV) (voltage range: 0 to 0.8 V) and galvanostatic charge/discharge (GCD) analyses were conducted under the same conditions as the EOT process. Moreover, a two-electrode system featuring symmetrical CNT and EOT CNT electrodes sandwiching a quasi-solid-state PVA-Na_2_SO_4_ electrolyte was constructed to test the applicability of the CNT sheet superconductor for bendable wearable devices. The gel electrolyte for both the CNT and EOT CNT sheets was produced by dissolving 6 g of PVA and 6 g of Na_2_SO_4_ in 60 mL of deionized water. This mixture was then stirred at 200 rpm and heated to 90 °C until it became clear.

### 2.3. Characterization

A cell phone camera (iPhone 11, Apple, California, USA) was used to obtain images of the CNT and EOT CNT sheets during bending. Scanning electron microscopy (SEM) images of the CNT and EOT CNT sheet supercapacitors were captured using an S-4600 instrument (Hitachi, Tokyo, Japan). Their electrochemical performances were evaluated using an electrochemical analyzer (Vertex EIS, Ivium, Noord-Brabant, The Netherlands). In addition, multi-meter probes (Model 187, Fluke Corporation, Washington, USA) were employed to measure resistance. Fourier-transform infrared spectroscopy (FTIR) was performed by using an IdentifyIR instrument (Smiths Detection, England). Raman spectroscopy (NRS-3100, JASCO, Tokyo, Japan) and X-ray photoelectron spectroscopy (XPS; ESCALAB 250XI, ThermoFisher Scientific, Waltham, MA, USA) were conducted to analyze the chemical characteristics of the CNT and EOT CNT sheets.

### 2.4. Electrochemical Performance Calculation

The areal capacitance (μF/cm^2^) was calculated from a CV curve using the following equation:(1)$$Capacitance=\frac{I/(dV/dt)}{unit}$$
where *I* is the average current of the CNT sheets, and *dV*/*dt* is the scan rate during the CV analysis.

## 3. Results and Discussion

### 3.1. Full Electrochemical Activation of the EOT CNT Sheet Electrode

The experimental setup for using EOT to functionalize the surface of CNT sheets (1 cm wide) is illustrated in Figure 1a. The working electrode was prepared by mechanically drawing and stacking five layers of the CNT sheets on a PET substrate (Figure 1b). This was used along with Ag/AgCl and Pt mesh as the reference and counter electrodes, respectively, in a three-electrode system. The CNT working electrode was immersed in an aqueous electrolyte (0.1 M Na_2_SO_4_) to electrochemically activate the CNT sheets and to characterize the electrochemical performance of the CNT electrode. The current density and resistance of the CNT sheets were measured during the EOT process (Figure 1c). It should be noted that the current density was approximately maintained during the initial phase of the EOT process but abruptly decreased after a certain treatment time. This point of current density drop coincides with a dramatic increase in the resistance of the CNT sheets, which could be due to excessive functionalization via oxygen-containing groups. Therefore, we defined just before the abrupt increase in resistance as the activation time used in all subsequent specimen preparations. Since the activation time varies depending on the applied voltage, the current densities according to time at voltages ranging from 2 to 3.5 V are shown in Figure 1d; the higher the voltage, the shorter the activation time. Moreover, the current density, activation time, and the resulting power and energy values according to the applied voltage are depicted in Figure 1e. The total energy required to activate the CNT sheets was similar at the various current densities and voltages (approximately 0.5 J), which infers much better cost and energy efficiency compared to previous oxidation methods such as sputtering. The SEM images before and after EOT, shown in Figure 1f and Figure 1g, respectively, reveal that EOT did not degrade the structural stability of the CNT sheets, resulting in high retention of the mechanical, electrical, and electrochemical properties of the electrodes. Moreover, the SEM image in Appendix A shows the anisotropic alignment of the CNT bundles after the EOT process.

### 3.2. Electrochemical Performance Measurements

One of the most compelling benefits of applying EOT to CNT sheet-based supercapacitors is its profound enhancement of the electrochemical performance compared to untreated CNT sheets. To quantify this effect, CV analysis was conducted over a voltage range from 0 to 0.8 V versus the Ag/AgCl reference electrode in a three-electrode system; CV curves for applied voltages of 2–3 V and oxidation times up to 40 s are shown in Figure 2a–c, respectively. Before EOT, the curves typically display a limited area indicative of low capacitance. However, after EOT, the area under the CV curves was dramatically enlarged, with growth roughly proportional to the treatment time. This infers a dramatic increase in charge storage capability due to EOT, which was confirmed through GCD analysis. The triangular GCD curves in Figure 2d signify an optimal capacitive charge storage mechanism while lengthening the charge/discharge time revealed capacitance augmentation via EOT. Although the main EOT parameters are applied voltage and treatment time, the influence of EOT on enhancing the capacitance seems to converge to a similar efficiency after a certain energy threshold was reached, yielding approximately a 40–50 times increase in the CV curve area (Figure 2e). Our subsequent analysis using a two-electrode system featuring symmetrical EOT CNT electrodes sandwiching a PVA-Na_2_SO_4_ quasi-solid-state electrolyte was conducted to validate the reliability of the capacitance improvement by the EOT effect. To this end, we performed multiple EOT experiments and plotted the resulting capacitance improvements and specific capacitances of 12 supercapacitors (Figure 2f). We also assessed the capacitance retention performance and high-rate capability of the pristine and EOT CNT sheet supercapacitors at various scan rates (from 10 to 1000 mV/s), of which the resulting areal capacitances are exhibited in Figure 2g; those of pristine and EOT CNT sheets measured at a scan rate of 10 mV/s were 18.2 and 587 μF/cm^2^, respectively. Moreover, 76% of the capacitance was retained at a very high scan rate of 1000 mV/s rate, as indicated by the well-defined rectangular CV curve at high scan rates without a notable loss in the CV curve area in Figure 2h. Finally, long-term stability was assessed through 1000 repeated charge/discharge cycles (Figure 2i), during which the EOT CNT sheet supercapacitor retained 89% of its performance. We also compared the areal capacitance of the EOT CNT sheets with those of previously reported carbon-based materials used as supercapacitor electrodes in Appendix A [36,37,38,39,40,41,42,43,44,45].

### 3.3. The Capacitance Improvement Mechanism

The FTIR spectra in Figure 3a offer definitive evidence of certain functional groups. The pronounced absorption peak in the spectrum of EOT CNTs near 3400 cm^−1^ arises from O-H stretching vibrations attributed to hydroxyl groups (specifically O=C-H and C-OH) stemming from the EOT of the CNT layers. The resonance near 1620 cm^−1^ is due to the vibrational stretching of C=C bonds. The peak at around 2330 cm^−1^ caused by the vibrational absorption of the C-O-C bonds was higher for the EOT CNTs, further verifying the oxidation of CNTs by EOT. The absorption peak around 1735 cm^−1^ linked to the carbonyl (C=O) group hints at the stretching vibrations of carboxyl units (-COOH). The resonance at around 1100 cm^−1^ related to the C-O stretching of the carboxylic acid group was higher in the spectrum of the EOT CNTs. These results indicate the successful formation of functional groups on the surface of the EOT CNTs. Raman spectroscopy provided additional evidence of functional group formation (Figure 3b). EOT increased the intensity of the D-band located at 1350 cm^−1^ to the G-band (located at 1590 cm^−1^ (*I_D_*/*I_G_*) ratio from 0.51 to 0.76, suggesting a higher defect density in the graphitic carbon structure and corroborating the FTIR findings. From our previous work [33], the hydroxyl group (C-OH) content seems to be pivotal for activating CNT.

XPS analysis was conducted to measure the degree of oxidation caused by EOT, with XPS spectra before and after EOT being presented in Appendix A, respectively. We found that the main functional groups present were hydroxyl and epoxy. The ratio of oxygen to carbon atoms (O/C) before and after EOT increased from 8% to 28%, respectively. This 20% increase is quite mild because EOT was terminated before the drastic increase in resistance in the CNT electrode (Appendix A), which could be due to excessive oxidation. If we apply harsh conditions by over-wetting the mild-oxidation region, the resistance drastically increases due to severe oxidation. Therefore, the internal resistance drop during the charge storage process predominates, resulting in the highly degraded CV curve shown in Appendix A.

Functional groups significantly improve the wettability of the CNT surface, thereby increasing the electrochemically active area for ion adsorption. To validate the improved wettability of the EOT CNT sheets over untreated ones, we analyzed the contact angles before and after EOT (Figure 3c and Figure 3d, respectively). The contact angle of water droplets on CNT sheets should be significantly different before and after EOT. Untreated CNT sheets generally exhibit a higher contact angle of approximately 125°, thereby indicating hydrophobicity due to a lack of polar functional groups on the surface (Figure 3c). On the other hand, the EOT CNT sheets had a considerably lower contact angle of approximately 55°, thereby indicating hydrophilicity and enhanced wettability (Figure 3d). Hence, the formation of oxygen-containing functional groups during EOT increased the surface polarity and consequently boosted wettability. A more hydrophilic surface enhances ion adsorption, leading to improved electrochemical performance. Therefore, the reduction in contact angle induced by EOT not only confirms the successful introduction of functional groups but also indicates an improvement in the electrochemical properties of the CNT sheets. This change in contact angle can thus act as a simple yet effective marker for the success of the EOT process in enhancing the performance of CNT-based supercapacitors.

### 3.4. Flexibility of the EOT CNT Sheet Supercapacitor

One of the standout features of our EOT CNT sheet-based supercapacitor is its exceptional mechanical flexibility, a property that is becoming increasingly important for wearable electronics, flexible displays, and portable energy storage devices [17]. To rigorously assess the flexibility of our supercapacitors, we fabricated elongated EOT CNT sheets (1 × 5 cm) and conducted a series of tests measuring resistance and discharge retention performances against bending curvature and the number of bending cycles. Our results demonstrate remarkable stability in both resistance and discharge retention under various bending conditions. Specifically, changes in the resistance of the supercapacitor were negligible across a range of bending curvatures (up to 0.63/cm), indicating that the electrical pathways within the CNT sheets remained largely intact, even when being bent to 180° (Figure 4a,b). Furthermore, the charge/discharge retention performance of the supercapacitor remained exceptionally stable under mechanical bending (the GCD curves measured statically while bending at 90° and 180° are shown in Figure 4c). Moreover, the dynamic charge/discharge characteristics analyzed via GCD and CV are shown in Figure 4d and Figure 4e, respectively. From these results, it is evident that the electrochemical performance of the EOT CNTs was well preserved while being bent. Typically, mechanical stress can result in reduced electrochemical performance due to disruptions in the active material or electrical contacts. However, our EOT CNT-based supercapacitor maintained a stable discharge profile even after thousands of bending cycles, thereby underscoring its robustness and reliability (Figure 4f and Appendix A). These findings not only affirm the mechanical flexibility of our CNT sheet-based supercapacitor but also highlight its potential for integration into a wide array of flexible and wearable devices.

## 4. Conclusions

We presented a groundbreaking approach to enhance the electrochemical performance of CNT sheet-based supercapacitors using EOT. Our work addresses the critical limitation of low capacitance found in pristine CNT sheets by introducing oxygen-containing functional groups onto the CNT surface via EOT. This approach improved ion adsorption by making the CNT sheets hydrophilic (a contact angle decrease from 125° to 55°), which consequently improved the capacitance by 54-fold. Moreover, the EOT CNT sheets retained their intrinsic mechanical and electrical properties, including exceptional flexibility and 98% capacitance retention after 1000 bending cycles. This represents a significant advancement in the field of energy storage, particularly for applications requiring flexible and durable supercapacitors. Our findings have far-reaching implications, not only in energy storage but also in wearable electronics, flexible displays, and other next-generation technologies.

## Figures and Tables

**Figure 1 nanomaterials-13-02814-f001:**
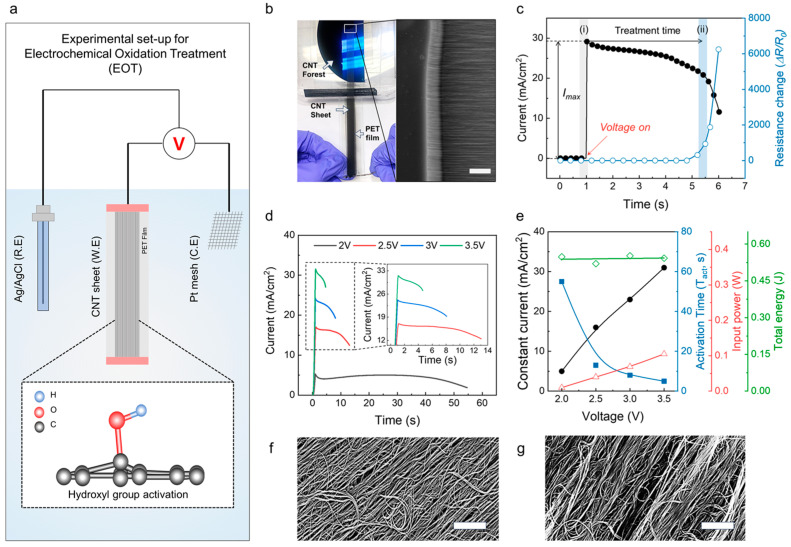
(**a**) A schematic illustration of the EOT setup including the CNT sheets (working electrode), Ag/AgCl (reference electrode), and Pt mesh (counter electrode) immersed in a 0.1 M Na_2_SO_4_ liquid electrolyte (inset: a schematic showing the expected functionalization of hydroxyl groups on the adjacent CNT surface). (**b**) A photograph showing the CNT sheet electrode preparation process. The aligned CNT sheets were mechanically drawn from the CNT forest and stacked on a PET film (scale bar = 40 mm). (**c**) Current density and sheet resistance versus oxidation time at a constant voltage of 3.5 V (versus Ag/AgCl). When the current plateaus before a sudden increase in resistance is defined as the activation time. ((i): treatment start region, (ii): treatment end region) (**d**) Current density versus oxidation time under various voltages (2, 2.5, 3, and 3.5 V) (versus Ag/AgCl). (**e**) Current density, treatment time, input power, and total energy versus the applied voltage. SEM images (**f**) before and (**g**) after EOT (scale bars = 1 μm).

**Figure 2 nanomaterials-13-02814-f002:**
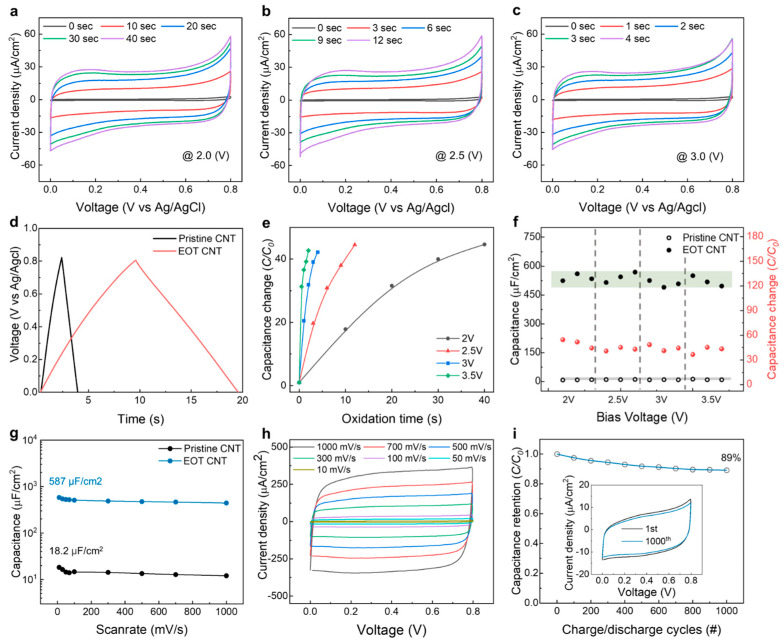
CV curves of EOT CNT sheets prepared at various activation times under applied voltages of (**a**) 2.0, (**b**) 2.5, or (**c**) 3.0 V at 30 mV/s. (**d**) GCD curves at a current density = 30 µA/cm^2^ of CNT sheets before and after EOT. (**e**) Capacitance change ratio versus oxidation time under applied voltages of 2, 2.5, 3, or 3.5 V. (**f**) Areal capacitance and capacitance change ratio versus applied voltage for 12 different supercapacitors in a two-electrode system consisting of two symmetrical electrodes coated with PVA/Na_2_SO_4_ gel electrolyte. (**g**) Areal capacitances of pristine and EOT CNT supercapacitors at scan rates from 10 to 1000 mV/s. (**h**) CV curves of the EOT CNT supercapacitor measured at scan rates from 10 to 1000 mV/s. (**i**) Capacitance retention versus charge/discharge cycles of the EOT CNT supercapacitor showing 89% retention after 1000 cycles (inset: the CV curves between the first and 1000th cycles at 30 mV/s).

**Figure 3 nanomaterials-13-02814-f003:**
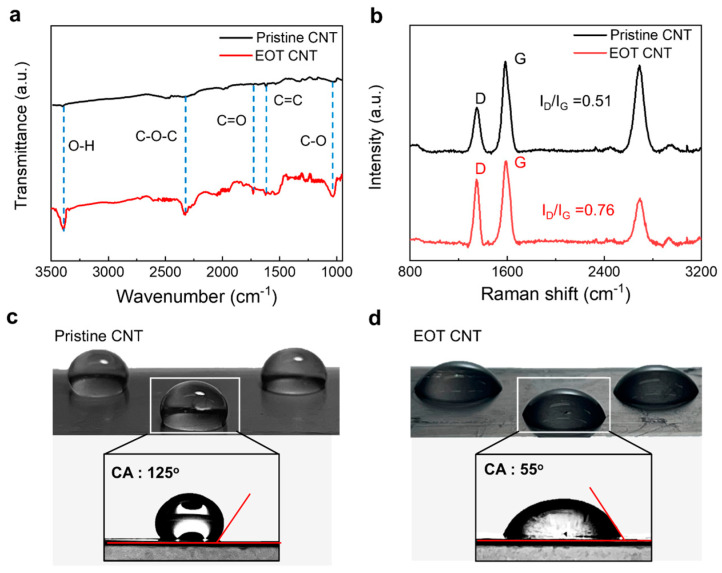
(**a**) FTIR and (**b**) Raman spectra of pristine and EOT CNT sheets(D: defect, G: graphite). (**c**) Optical images showing water droplets and their contact angles (CAs) on (**c**) pristine CNT sheets and (**d**) EOT CNT sheets.

**Figure 4 nanomaterials-13-02814-f004:**
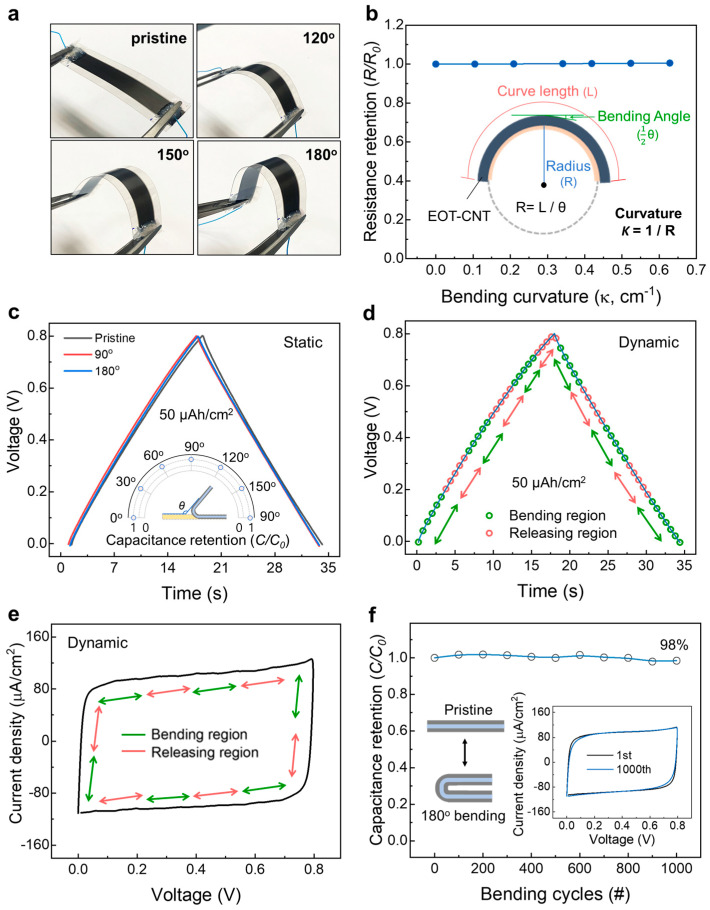
(**a**) Optical images showing an EOT CNT sheet electrode bent at various angles. (**b**) Resistance change ratio versus bending curvature of EOT CNT sheets at 0–180°. (**c**) A GCD curve at a current density of 50 µA/cm^2^ of the bent EOT CNT sheets. The test was performed with a two-electrode system while being bent at 0°, 90°, or 180°. (**d**) GCD curves at a current density of 50 µA/cm^2^ and (**e**) CV curves at 30 mV/s measured during 180° bending and releasing dynamic cycles of the EOT CNT sheets. (**f**) Capacitance retention versus 180° bending cycles of the EOT CNT sheets showing 98% capacitance retention after 1000 cycles (inset: a comparison of the CV curves between the first and 1000th bending cycles at 30 mV/s).

## Data Availability

Not applicable.

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
