# Peer review of "Electrochemically Oxidized Carbon Nanotube Sheets for High-Performance and Flexible-Film Supercapacitors"

_nanomaterials, 2023, doi:10.3390/nano13202814_

Round 1

Reviewer 1 Report

Minor edition is needed.

Author Response

We appreciate the comments of the reviewers and the suggestions of the editor. Our responses to these comments are listed below, and the manuscript and supplemental materials have been accordingly revised to clearly address the issues raised by reviewers. To help address the comments of the reviewers, we performed additional experiments and added pertinent new results. The main revision contents include (1) XPS analysis to characterize the functional groups introduced during the electrochemical wetting process, (2) SEM measurements to clearly demonstrate the structural stability of the CNT sheets after the electrochemical wetting, and (3) rewriting the abstract, introduction, conclusion, and experimental parts to more clearly explain our results. We performed our best to address the issues raised by reviewers in the limited revision due time. According to the comments from the reviewer, the manuscript was revised using more abundant expressions and sentence patterns. In addition, the English in the revised manuscript was edited by an editing firm (essay review). 

Reviewer 2 Report

In this manuscript, aiming the high performance energy storage, authors developed oxidized carbon nanotube sheets for the flexible film supercapacitors. Comprehensive characterizations have been performed and the work is interesting. In general, the manuscript is well organized. However, there are still some issues to be addressed. A moderate revision is suggested before its acceptance.

1.     More solid data should be included in abstract section.

2.     It is better to use full name in keywords.

3.     In introduction, authors are suggested to provide comparison to other carbon based materials for supercapacitor electrode to present the novelty of using CNTs in this work.

4.     The last paragraph in introduction can be further introduced the novelty, strategy, method and important results.

5.     The generally introduction of the different energy storage sources should be provided with some more recent supporting articles, such as Li-ion battery (Journal of Energy Storage, 72 (2023) 108509); aqueous Zn-ion batteries (J Alloys  Compounds, 2022, 903: 163824); lithium–selenium batteries (Rare Met, 2022, 41(10): 3432-3445); aqueous ammonium-ion batteries (Chemical Engineering Journal, 2023, 458, 141381); Li-S batteries (Adv. Energy Mater. https://doi.org/10.1002/aenm.202302139); Zn-air battery (Molecules 28 (5), 2147, 2023); supercapacitor (Chinese Chemical Letters 109007, https://doi.org/10.1016/j.cclet.2023.109); etc.

6.     One sub-section on the raw materials can be provided.

7.     One scheme to show the experimental procedure is suggested for better understanding of this work to readers.

8.     Some figures should be modified with a better readability, especially the quite small texts.

9.     Authors only applied 1000 cycles for the cycling test. More cycles should be applied to show the novelty of this work.

10.  Authors should recheck the references to make sure full information is provided, such as volume, pages, etc. In addition, the format of references should be uniform.

11.  There are still some typos and grammar issues in the manuscript. Authors should carefully recheck the whole manuscript.

Author Response

(The authors gave the same response as above.)

Reviewer 3 Report

The MS described  the integration of EOT with CNT sheets, achieving improved electrochemical performance and mechanical flexibility.   The MS can be accepted for the publication, after the following points are considered:

1. SEM images are not clear - from those shown in the MS nothing can be seen;

2. The Conclusions are not well written. Conclusions must state the improvements brought by the work (not the originality), and also the features, and the possible applications.

Author Response

(The authors gave the same response as above.)

Reviewer 4 Report

This manuscript is about the "Electrochemically Oxidized Carbon Nanotube Sheets for 2 Highly Performing and Flexible Film Supercapacitors". The work is interesting, and the capacitance and retention are acceptable. Therefore, I would like to recommend the publication of this work after major revisions.

My comments below:

#Abstract requires more technical achievements from the proposed works compared to highlighting the novelty of the manuscript.

#No details about the experimental measurements were provided in part 2.2, only the names not models of the instruments.

#What about the crystalline phase of tubes? The author should perform XRD.

#Why does the ID/IG ratio suggest the presence of micropores in the tubes?

#Why are the SEM images not clear? The authors should provides different magnification images (Since CNT is a nanometer, provided in 20 nm, 50 nm, 100 nm, and 200 nm) to get a clear evaluation of the prepared material.

#What about the microstructures of CNT?

#The endurance tests should also be marked with the current density in the figure caption part.

#What about structural stability after the measurements?

Essentially related work should be compared to present work with previously reported works, summarized in the comparison Table.

Authors did not explain the limitations of the work, should I assume that there are no limitations? It would be nice if they said the future perspectives and its limitations which can attract more readers.

This manuscript contains spelling typos, style errors, and grammatical errors, which severely affect readability. Please carefully check the whole manuscript and correct it.

Minor editing of English language required

Author Response

(The authors gave the same response as above.)

Round 2

Reviewer 1 Report

All of my concerns were well addressed, and the present version is acceptable for publication. 

minor editing

Reviewer 2 Report

Authors have made a well revision. An acceptance is suggested.

Reviewer 3 Report

The MS is ok now, and can be accepted for publication. 

Reviewer 4 Report

As the authors replied all the queries raised by the reviewers satisfactorily and modified the text accordingly. I strongly recommend that the revised manuscript can be considered for publication in the present form.